# Neocortical plasticity: an unsupervised cake but no free lunch

**Eilif B. Muller***

**Philippe Beaudoin**
Element AI
6650 Saint-Urbain #500
Montreal, QC H2S 3G9
Canada

*Correspondence to: eilif.muller@elementai.com

## Abstract

The fields of artificial intelligence and neuroscience have a long history of fertile bi-directional interactions. On the one hand, important inspiration for the development of artificial intelligence systems has come from the study of natural systems of intelligence, the mammalian neocortex in particular. On the other, important inspiration for models and theories of the brain have emerged from artificial intelligence research. A central question at the intersection of these two areas is concerned with the processes by which neocortex learns, and the extent to which they are analogous to the back-propagation training algorithm of deep networks. Matching the data efficiency, transfer and generalization properties of neocortical learning remains an area of active research in the field of deep learning. Recent advances in our understanding of neuronal, synaptic and dendritic physiology of the neocortex suggest new approaches for unsupervised representation learning, perhaps through a new class of objective functions, which could act alongside or in lieu of back-propagation. Such local learning rules have implicit rather than explicit objectives with respect to the training data, facilitating domain adaptation and generalization. Incorporating them into deep networks for representation learning could better leverage unlabelled datasets to offer significant improvements in data efficiency of downstream supervised readout learning, and reduce susceptibility to adversarial perturbations, at the cost of a more restricted domain of applicability.

## Unsupervised neocortex

The neocortex is the canonically 6-layered sheet of cells forming the grey matter surface of the mammalian cerebrum. It is composed of a densely interconnected network of sub-regions responsible for learning sensory processing, speech and language, motor planning and many of the higher cognitive processes associated with rational thought. The human neocortex contains an estimated 100 trillion synapses, the points of communication between neurons which undergo persistent changes in strength and topology as a function of signals local to the synapse and a complex biochemical program (Holtmaat and Svoboda, 2009). These processes, broadly known as synaptic plasticity, are thought to be the basis of learning and memory in the brain.

An important task of synaptic plasticity in sensory neocortical areas is to learn disentangled invariant representations (DiCarlo et al., 2012). For example, the ventral stream of primate visual cortex,

33rd Conference on Neural Information Processing Systems (NeurIPS 2019), Vancouver, Canada.

the collection of areas responsible for visual object recognition, computes hierarchically organized representations much like state-of-the art convolutional neural networks (CNNs) optimized for the task (Yamins et al., 2014).

While there are impressive similarities in the learned representations between the ventral stream and CNNs, there are important differences in *how* those representations are learned. While CNNs are trained in a supervised manner using a gradient descent optimization algorithm with an explicit global objective on large labelled datasets, the ventral stream learns from a much larger dataset (visual experience) but with only very sparse labelling. The latter property of cortical learning is attractive to emulate in CNNs, and more broadly across deep learning models. Attractive, not only because of the ability to make use of unlabelled data during learning, but also because it will impart the models with superior generalization and transfer properties, as discussed below.

## The monkey's paw effect: the problem with specifying what without specifying how

A well known and often encountered pitfall of numerical optimization algorithms for high dimensional problems, such as evolutionary algorithms, simulated annealing and also gradient descent, is that they regularly yield solutions matching *what* your objective specifies to the letter, but far from *how* you intended (Lehman et al., 2018).

The short story "The Monkey's Paw" by W. W. Jacobs provides a compelling metaphor. In that story, the new owner of a magical mummified monkey's paw of Indian origin is granted three wishes. The owner first wishes for $200, and his wish is eventually granted to the penny, but with the grave side effect that it is granted through a goodwill payment from his son's employer in response to his untimely death in a terrible machinery accident (Jacobs and Parker, 1910).

The Monkey's Paw effect is also applicable to gradient descent-based optimization of deep neural nets. The relative data-hungriness of current supervised learning strategies, and the use of data augmentation to improve generalization reflect the precarious position we are in of needing to micromanage the learning processes.

Adversarial examples (Moosavi-Dezfooli et al., 2016) are evidence that the monkey's paw effect none-the-less persists. It is temping to continue with the current paradigm and re-inject adversarial examples back into the learning data stream. Extrapolating, this goes in the direction of specifying the negative space of the objective, all those things the optimization should not do to solve the problem, which is potentially infinite, and rather risky in production environments like self-driving cars.

Adversarial examples represent an opportunity to address the issue in a more fundamental way (Yamins and DiCarlo, 2016). It has been argued by Bengio (2012) that if we could design deep learning systems with the explicit objective of "disentangling the underlying factors of variation" in an unsupervised manner, then there is much to be gained for generalization and transfer.

Such an approach offers a promising solution to the Monkey's Paw effect, as there is an explicit objective of learning good representations, from which generalization and transfer follow by definition.[1] One small challenge remains: how to express the objective of learning good representations? If we restrict ourselves to the subset of all possible inputs for which the neocortex learns good representations, the local processes of synaptic plasticity may provide valuable clues.

## Neocortical plasticity

The neocognitron model (Fukushima, 1980), the original CNN architecture, learned visual features through self-organization using local rules. Since its conception, our understanding of the neocortex and its neurons and synapses has progressed considerably.

Recent insights into the local plasticity rules for learning in the neocortex offer new inspiration for deep representation learning paradigms that learn "disentangled representations" from large unlabelled datasets in an unsupervised manner. A selection of recent insights into the systems of plasticity of the neocortex is shown in Fig. 1. A new dendrite-centric view of synaptic plasticity is

---

[1]For some input spaces, such as white noise, a good representation may be undefined.

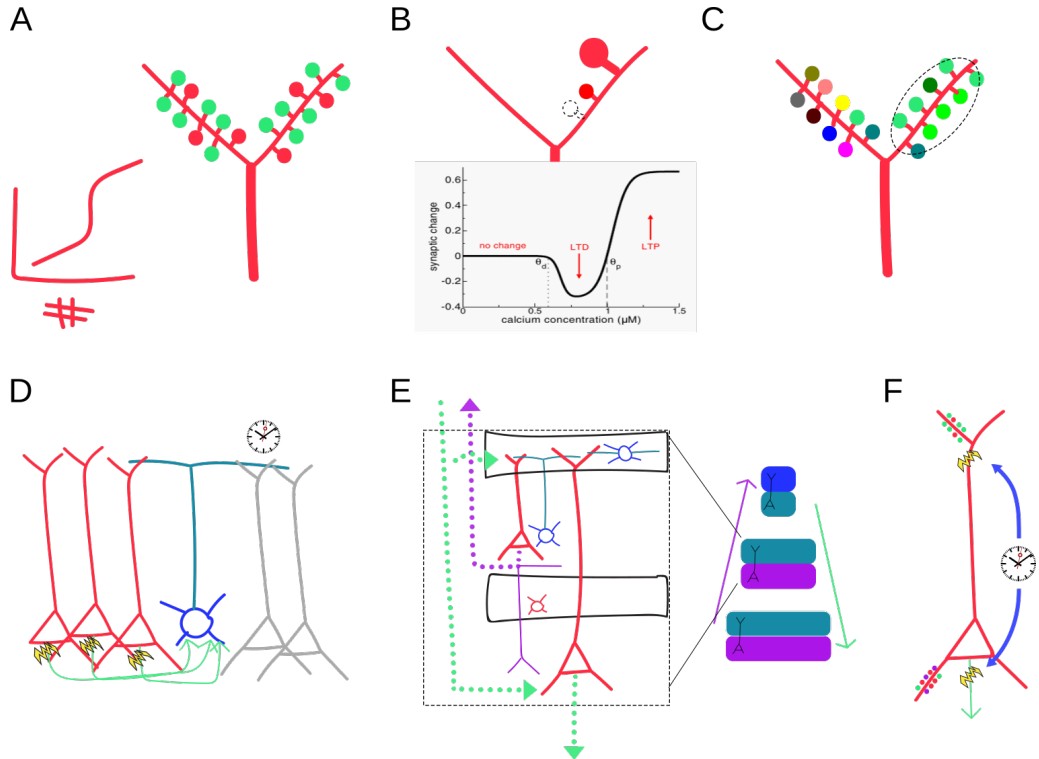

Figure 1: **A selection of recent insights into the dendritic mechanisms of plasticity of the neocortex.** (**A**) Concurrent activation of > 10 nearby synapses in pyramidal neuron dendrites (red) triggers NMDA plateau potentials in dendrites (left). (**B**) Calcium drives synaptic plasticity. Synapses are bi-stable, and can be added or removed in the weak state (above). NMDA plateau potentials drive potentiation of synapses through their associated large calcium currents. (source: Graupner and Brunel (2010)) (**C**): Clusters of co-coding synapses are captured through these mechanisms. (**D**) Co-coding neurons form small cliques, reinforced through cluster capture. These cliques activate Martinotti cells which block further capture, implementing opposing competition. (**E**) Neocortical areas are organized in a hierarchy with top-down input arriving in layer 1 (the top-most layer) at the apical tufts of pyramidal dendrites, and at layer 6 and lower layer 5. (**F**). Temporal association of top-down and bottom-up drives cliques and plasticity.

emerging with the discovery of the NMDA spike, a non-linear mechanism hypothesized to associate co-activated synapses through potentiation or structural changes driven by the resulting calcium currents (Schiller et al., 2000; Graupner and Brunel, 2010; Holtmaat and Svoboda, 2009) (Fig. 1A-B). Such associations, in the form of co-coding clusters of synapses, have recently been experimentally observed using optical techniques (Wilson et al., 2016) (Fig. 1C). Moreover neurons in the neocortex are known to form small cliques of all-to-all connected neurons which drive co-coding (Reimann et al., 2017), a process that would be self-reinforced through dendritic clustering by NMDA spikes (Fig. 1D). Martinotti neurons, which are activated by such cliques of pyramidal neurons, and subsequently inhibit pyramidal dendrites (Silberberg and Markram, 2007) provide well-timed inhibition to block further NMDA spikes (Doron et al., 2017), and put a limit on the maximal pyramidal clique size, but also suppress activation of competing cliques (e.g. Winner-take-all (WTA) dynamics). Together, such plasticity mechanisms appear to form basic building blocks for representation learning in the feed-forward pathway of the neocortex using local learning rules. While long known competitive strategies for unsupervised representation learning indeed rely on WTA dynamics (Fukushima, 1980; Rumelhart and Zipser, 1985), deep learning approaches incorporating these increasingly apparent dendritic dimensions of learning processes have yet to be proposed (Poirazi and Mel, 2001; Kastellakis et al., 2015).

Unlike CNNs, the neocortex also has a prominent feedback pathway down the hierarchy, whereby top-down input from upper layers innervate the apical tufts of pyramidal cells in layer 1 of a given cortical

region (Felleman and Van, 1991). Associations between top-down and feed-forward (bottom-up) activation are known to trigger dendritic calcium spikes and dendritic bursting (Larkum et al., 1999), which again specifically activates the WTA dynamics of the Martinotti neurons (Murayama et al., 2009), but disinhibitory VIP neurons can also modulate their impact (Karnani et al., 2016). These feed-back pathways have been proposed to implement *predictive coding* (Rao and Ballard, 1999), and error back-propagation for supervised learning algorithms (Guerguiev et al., 2017; Sacramento et al., 2018). While their importance for rapid object recognition has been recently demonstrated, their computational role remained inconclusive (Kar et al., 2019).

## Cake but no free lunch

With the demonstrated applicability of supervised learning for a broad range of problems and data distributions, and an ever expanding toolbox of optimized software libraries, it is unlikely that supervised learning, back-propagation and gradient descent will be dethroned as the work horses of AI for many years to come.

Nonetheless, as applications of deep networks are moving into regions where sparse data, generalization and transfer are increasingly important, unsupervised approaches designed with the explicit goal of learning good representations from mere observation may find an important place in the AI ecosystem.

Quoting Yann LeCun[2]

> "If intelligence is a cake, the bulk of the cake is unsupervised learning, the icing on the cake is supervised learning, and the cherry on the cake is reinforcement learning."

A promising strategy would be to assume learning with sparse labels, overcoming adversarial examples, transfer learning, and few-shot learning together as the success criteria for the further development of the powerful unsupervised approaches we seek.

Recent advances in our understanding of the processes of neocortical plasticity may well offer useful inspiration, but let's close with some words of moderation. Biology's solutions also show us there will be no free lunch, i.e. neocortical unsupervised learning algorithms will be less general than supervised learning by gradient descent. Neocortex relies on structure at specific spatial and temporal scales in its input streams to learn representations. Evolution has had millions of years to configure the sensory organs to provide signals to the neocortex in ways that it can make sense of them, and that serve the animal's ecological niche. We should not expect, for example, cortical unsupervised learning algorithms to cluster frozen white noise images. A neocortical solution requires a neocortical problem (e.g. from the so-called "Brain set" (Richards et al., 2019)), so if we are to successfully take inspiration from it, we must also work within its limitations.

### Acknowledgments

Thanks to Giuseppe Chindemi, Perouz Taslakian, Pau Rodriguez, Isabeau Prémont-Schwarz, Hector Palacios Verdes, Pierre-André Noël, Nicolas Chapados, Blake Richards, and Guillaume Lajoie for helpful discussions.

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
