# OpenReview forum: "Neocortical plasticity: an unsupervised cake but no free lunch"
_NeurIPS.cc/2019/Workshop/Neuro_AI — Real Neurons & Hidden Units @ NeurIPS 2019 Poster_

### Official Review · AnonReviewer1 · 2019-09-17
**Discussion of various ideas in neuroscience and machine learning, but details largely absent**

**Clarity:** 2

**Comment:**

I would challenge the authors to expand the discussion around Martinotti neurons to include a model of how such mechanisms could facilitate unsupervised / representation learning.

The figure could be improved by converting the hand-drawn diagrams into more professional looking graphics.

The introduction contains several overstated claims. For instance, it’s difficult to say whether the “task” of synaptic plasticity in the neocortex is to learn disentangled representations.

**Category:**

Neuro->AI

**Clarity Comment:**

The submission was, at times, difficult to follow. For instance, connections between the neuroscience discussion (e.g. cliques of pyramidal neurons) and better forms of representation learning are unclear.

**Evaluation:**

2: Poor

**Importance:**

2: Marginally important

**Importance Comment:**

The submission discusses a few observations about the neocortex and emphasizes the importance of unsupervised learning of representations. However, the connections between these points are unclear, and it is therefore difficult to determine if any novel ideas are proposed. Because the submission does not elaborate on how neocortical principles could assist in improving unsupervised representation learning, the importance of this work seems lacking.

**Intersection:**

4: High

**Intersection Comment:**

The submission discusses both fields. However, as mentioned, the details in connecting these areas are largely absent.

**Rigor Comment:**

The submission does not present any empirical results or theoretical formulations. Technical concepts are not explored in detail.

**Technical Rigor:**

2: Marginally convincing

---

### Official Review · AnonReviewer3 · 2019-09-20
**An interesting proposal, but still very preliminary/loose**

**Clarity:** 5

**Comment:**

There are some great ideas in here, and potentially excellent topics for discussion. But, there is very little in terms of actual material contributions. It is essentially an opinion piece.

**Category:**

Neuro->AI

**Clarity Comment:**

It's a very well written submission.

**Evaluation:**

3: Good

**Importance:**

4: Very important

**Importance Comment:**

The importance of understanding unsupervised learning in the brain cannot be understated. If we could emulate the unsupervised learning used by the brain in ANNs it would be a massive leap forward in AI. Thus, the goals of this submission are very important. However, this submission only gestures at potential solutions, so the importance of this specific contribution is more limited.

**Intersection:**

5: Outstanding

**Intersection Comment:**

It is a perfect mix of neuroscience and AI.

**Rigor Comment:**

This submission contains much speculation, and some discussion of known biological facts. But, there is no analytical or empirical demonstration that the biological mechanisms described actually would provide the sort of unsupervised learning proposed. Furthermore, the claim that such unsupervised mechanisms would prevent susceptibility to adversarial attacks is unconvincing, and not backed up by any data or math.

**Technical Rigor:**

1: Not convincing

---

### Official Review · AnonReviewer2 · 2019-09-20
**A critical topic for discussion at the intersection of neuroscience and AI, but few novel ideas proposed**

**Clarity:** 4

**Comment:**

I strongly believe the topics covered in this paper should be discussed in this workshop. However, I do not believe this paper stands to contribute much to such a discussion, as it provides few novel insights or directions. One somewhat novel idea that this reviewer was able to walk away with was the suggestion that, in order to obtain good models of biological learning, we should focus on solving ecologically relevant statistical problems in AI.

**Category:**

Common question to both AI & Neuro

**Clarity Comment:**

Generally well-written.

**Evaluation:**

2: Poor

**Importance:**

3: Important

**Importance Comment:**

The importance of the topic covered in this paper - namely, the role of unsupervised learning in biology and artificial intelligence - is very high. However, this point has been highlighted frequently in the past, and this paper stops short of offering any concrete or novel contributions.

**Intersection:**

5: Outstanding

**Intersection Comment:**

This topic surely should and will be discussed at this workshop.

**Rigor Comment:**

No concrete results or proposals are offered to solve the (important) problem of unsupervised learning in AI and the brain. Experimental findings regarding NMDA-mediated plasticity in cortex are briefly reviewed, but not connected back to this problem, providing little insight into how to solve it.

**Technical Rigor:**

1: Not convincing

---

### Decision · Program_Chairs · 2019-10-02

Accept (Poster)